# A rechargeable Ca/Cl$_2$ battery

Shitao Geng[1,4], Xiaoju Zhao[1,4], Qiuchen Xu[1], Bin Yuan[1], Yan Wang[1], Meng Liao[2], Lei Ye[2], Shuo Wang[1], Zhaofeng Ouyang [1], Liang Wu [1], Yongyang Wang [3], Chenyan Ma[3], Xiaojuan Zhao[3] & Hao Sun [1] ✉

Rechargeable calcium (Ca) metal batteries are promising candidates for sustainable energy storage due to the abundance of Ca in Earth's crust and the advantageous theoretical capacity and voltage of these batteries. However, the development of practical Ca metal batteries has been severely hampered by the current cathode chemistries, which limit the available energy and power densities, as well as their insufficient capacity retention and low-temperature capability. Here, we describe the rechargeable Ca/Cl$_2$ battery based on a reversible cathode redox reaction between CaCl$_2$ and Cl$_2$, which is enabled by the use of lithium difluoro(oxalate)borate as a key electrolyte mediator to facilitate the dissociation and distribution of Cl-based species and Ca$^{2+}$. Our rechargeable Ca/Cl$_2$ battery can deliver discharge voltages of 3 V and exhibits remarkable specific capacity (1000 mAh g$^{-1}$) and rate capability (500 mA g$^{-1}$). In addition, the excellent capacity retention (96.5% after 30 days) and low-temperature capability (down to 0 °C) allow us to overcome the long-standing bottleneck of rechargeable Ca metal batteries.

Modern electrification has witnessed the ever-growing demand for rechargeable batteries with high sustainability and energy storage capabilities[1–4]. Rechargeable calcium (Ca) metal batteries are among the most promising candidates because of their advantageous features, such as high crustal abundance, high theoretical capacity, and ideal redox potential[5–7]. However, compared with the conventional Li counterparts, current Ca metal batteries based on intercalation/deintercalation cathodes, such as metal oxides, suffer from relatively limited electrochemical performance, e.g., low specific capacities (<200 mAh g$^{-1}$), discharge voltages (<2.6 V), and rate capability (<100 mA g$^{-1}$)[8–11], which requires the development of new cathode reactions to overcome these limitations on electrochemical performance. In addition, the capacity retention and low-temperature performance of Ca metal batteries remain inaccessible to date, due to the inferior electrochemical stability and sluggish cathode/anode reaction kinetics. Therefore, it is crucial yet challenging to develop new cathode chemistry to realize practical applications of rechargeable Ca metal batteries[12–14].

Reversible CaCl$_2$/Cl$_2$ redox reaction is among the most attractive cathode reactions for Ca metal batteries, enabling the delivery of high

specific capacities (483 mAh g$^{-1}$ based on the mass of CaCl$_2$) at discharge voltages of 3 V. However, it is currently unavailable so far, owing to the challenging control of Ca-based electrolytes and the generated electrolyte/electrode interfaces[3]. On the cathode side, for instance, the strong electrostatic force of Ca$^{2+}$ is prone to inhibit its efficient dissociation and distribution in the electrolyte[15,16], making the desired CaCl$_2$/Cl$_2$ cathode conversion thermodynamically and kinetically unfavorable. On the anode side, the high reactivity of Ca metal requires rational regulation of electrolyte solvation to ensure high electrochemical reversibility[17,18]. Therefore, it is desired but challenging for electrolyte designing and screening to unlock reversible CaCl$_2$/Cl$_2$ redox reactions for practical Ca metal batteries.

Here, we show that reversible CaCl$_2$/Cl$_2$ redox reaction can be fully unlocked in Ca metal batteries using a CaCl$_2$–AlCl$_3$–SOCl$_2$ electrolyte regulated by a lithium difluoro(oxalate)borate (LiDFOB) mediator. LiDFOB has been demonstrated to facilitate the dissociation and distribution of Cl-based species and Ca$^{2+}$, thus allowing the CaCl$_2$/Cl$_2$ redox reaction. Our rechargeable Ca/Cl$_2$ battery demonstrates discharge voltages of 3 V and remarkable specific capacity

[1]Frontiers Science Center for Transformative Molecules, School of Chemistry and Chemical Engineering, and Zhangjiang Institute for Advanced Study, Shanghai Jiao Tong University, 200240 Shanghai, China. [2]Department of Mechanical Engineering, The Pennsylvania State University, University Park, State College, PA 16802, USA. [3]Beijing Synchrotron Radiation Facility (BSRF), Institute of High Energy Physics, Chinese Academy of Sciences, 100049 Beijing, China. [4]These authors contributed equally: Shitao Geng, Xiaoju Zhao. ✉e-mail: haosun@sjtu.edu.cn

(up to 1000 mAh g⁻¹) and rate capability (up to 500 mA g⁻¹) because of the fast kinetics of the $CaCl_2/Cl_2$ redox reaction. In addition, the shelf life (>30 days) and low-temperature performance (down to 0 °C) achieved by this battery overcome the long-standing bottlenecks that have plagued the practical application of rechargeable Ca metal batteries. Our results provide a new paradigm for bridging Cl-based cathode and multivalent metal anode chemistry to achieve sustainable and high-performance energy storage.

## Results

To develop a rechargeable $Ca/Cl_2$ battery, we used a graphite cathode and a Ca metal anode coupled with a Cl-based electrolyte composed of $CaCl_2$, $AlCl_3$, and LiDFOB salts in $SOCl_2$ (named CALS electrolyte) (Fig. 1a and Supplementary Fig. 1; see preparation details in "Methods"). This rechargeable $Ca/Cl_2$ battery delivered a high reversible specific capacity of 1000 mAh g⁻¹ and a high discharge voltage of more than 3 V (Fig. 1b), making it highly competitive compared with state-of-the-art cathodes used in Ca metal batteries, e.g., metal oxides[19], sulfur[20], and oxygen[21] (Fig. 1c and Supplementary Fig. 2).

Notably, $CaCl_2$ was in-situ formed on the graphite cathode during the first discharge of the battery, as shown in Eq. (1), and could stably exist at the cathode because the dissolution of $CaCl_2$ in the electrolyte has reached saturation. At the anode, the stripping of Ca metal occurred as described in Eq. (2). The overall reaction during the first discharge was described in Eq. (3). A specific capacity of 3,264 mAh g⁻¹ (based on the mass of graphite here and throughout this paper) was delivered with a discharge voltage of ~2.8 V at 100 mA g⁻¹ (Supplementary Fig. 3). The formation of $SO_2$ was verified

by differential electrochemical mass spectrometry (DEMS, Supplementary Fig. 4).

$$\text{Cathode}: 2SOCl_2 + 2Ca^{2+} + 4e^- \rightarrow 2CaCl_2 + S + SO_2 \quad (1)$$

$$\text{Anode}: 2Ca - 4e^- \rightarrow 2Ca^{2+} \quad (2)$$

$$\text{Overall}: 2Ca + 2SOCl_2 \rightarrow 2CaCl_2 + S + SO_2 \quad (3)$$

The formed $CaCl_2$ on the graphite cathode was then oxidized to $Cl_2$ during the subsequent charge step and reduced back to $CaCl_2$ during discharge according to Eq. (4), while the deposition and stripping process of Ca metal at the anode was shown in Eq. (5), resulting in a high discharge plateau at ~3.5 V (Fig. 1b). The overall reaction of the subsequent charging and discharging processes was shown in Eq. (6). A minor discharge plateau was observed at ~2 V, which might correspond to the reduction of other charge products, such as $SO_2Cl_2$[22].

$$\text{Cathode}: CaCl_2 - 2e^- \leftrightarrow Ca^{2+} + Cl_2 \quad (4)$$

$$\text{Anode}: Ca^{2+} + 2e^- \leftrightarrow Ca \quad (5)$$

$$\text{Overall}: CaCl_2 \leftrightarrow Ca + Cl_2 \quad (6)$$

We found that LiDFOB played a key role in enabling the rechargeability of the $Ca/Cl_2$ battery. For instance, the battery prepared with

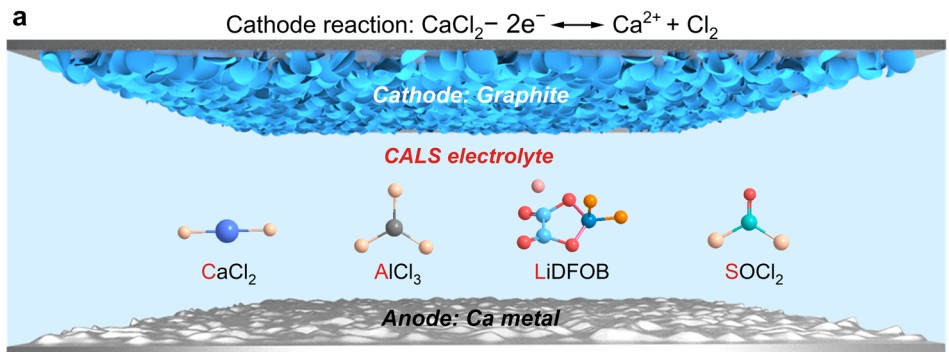

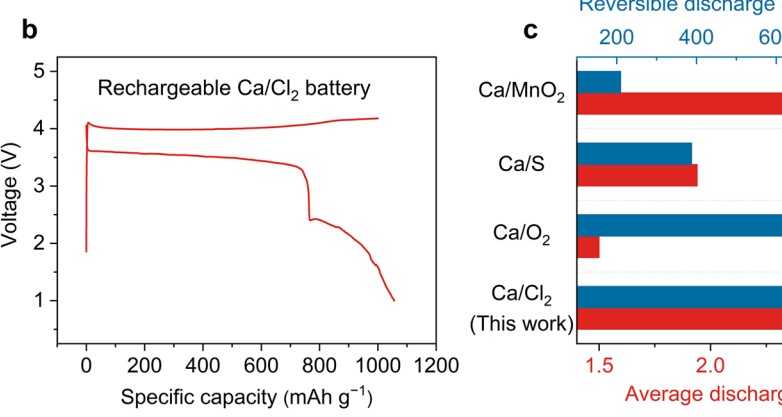

**Fig. 1 | Rechargeable $Ca/Cl_2$ battery based on a Cl-based electrolyte mediated by LiDFOB. a** Schematic illustration of a rechargeable $Ca/Cl_2$ battery based on a graphite cathode, a Ca metal anode, and a CALS electrolyte composed of a mixture of $CaCl_2$, $AlCl_3$, and LiDFOB dissolved in $SOCl_2$. **b** Galvanostatic charge–discharge curve of the rechargeable $Ca/Cl_2$ battery. The specific charge capacity and current density are 1000 mAh g⁻¹ and 100 mA g⁻¹, respectively. **c** Comparison of the

reversible specific capacity and discharge voltage of the rechargeable $Ca/Cl_2$ battery with those of other representative batteries with Ca cathodes, e.g., manganese dioxide[19], sulfur[20], and oxygen[21]. All specific capacities were based on the mass of the active materials on the cathode. A comparison of the areal capacities was also provided in Supplementary Fig. 2.

a LiDFOB-free electrolyte did not deliver any charge/discharge capacity, and no plateau was observed, in stark contrast to the pronounced charge/discharge behavior observed in the presence of LiDFOB (Fig. 2a), which inspired us to investigate the origin of the rechargeability of our Ca/Cl₂ battery. We hypothesized that the unique chemical structure of DFOB⁻ with two B−F and two C=O functional groups could effectively mediate the dissociation and distribution of Cl-based species (e.g., AlCl₃ and SOCl₂) and Ca²⁺, thus facilitating their immigration and improving the kinetics of the redox reaction (Fig. 2b). To test this hypothesis, we acquired Raman spectra of different electrolytes (Fig. 2c). The spectrum of bare SOCl₂ showed peaks attributed to the characteristic symmetric Cl−S−Cl deformation ($\delta_s$(Cl−S−Cl), 342.8 cm⁻¹), asymmetric S−Cl stretching ($\nu_a$(S−Cl), 441.1 cm⁻¹) and symmetric S−Cl stretching ($\nu_s$(S−Cl), 487.7 cm⁻¹) vibrations[23,24]. The addition of AlCl₃ resulted in two new peaks at 384.9 and 524.6 cm⁻¹, corresponding to the formation of Cl₂SO···AlCl₃ adducts. Subsequent addition of CaCl₂ showed two new Raman peaks at 363.8 and 504.5 cm⁻¹, which could be attributed to the Ca−O vibration (Ca²⁺···SOCl₂). In addition, the coordination peaks of ν(Al−O) and ν(S−Cl) were redshifted and weakened, indicating a weakened interaction between AlCl₃ and SOCl₂. Further introduction of DFOB⁻ significantly changed the Raman profile. On the one hand, the Al−O stretching vibration peak significantly redshifted from 382.6 to 376.9 cm⁻¹ with a significant intensity decrease. On the other hand,

the coordination S−Cl stretching (AlCl₃···SOCl₂) peak at 524.6 cm⁻¹ almost disappeared, confirming the sufficient dissociation between AlCl₃ and SOCl₂ mediated by DFOB⁻. This evidence indicated that the introduction of DFOB⁻ could effectively facilitate the dissociation of the strong interaction between AlCl₃ and SOCl₂, thus enabling reversible and fast CaCl₂/Cl₂ redox reaction. In addition, the solvation structure of the CALS electrolyte remained almost unchanged before and after gelation, as verified by the highly consistent Raman spectra in Supplementary Fig. 5. Thermogravimetric analysis and differential scanning calorimetry tests indicate that the gelation of CALS electrolyte may be related to the intermolecular interaction between different substances such as Ca²⁺ and DFOB⁻ (Supplementary Fig. 6)[25,26].

The distribution and immigration of Ca²⁺ were also significantly improved with the introduction of DFOB⁻. We performed molecular dynamics (MD) simulations to investigate the solvation structure of Ca²⁺ mediated by DFOB⁻ (Fig. 2d−g; see details in the Methods of Supplementary Information), which was further confirmed by the Ca K-edge X-ray absorption spectra (XAS) (Supplementary Fig. 7). In the absence of DFOB⁻, the solvation shell of Ca²⁺ ions was only composed of Cl-based species such as Cl⁻ and AlCl₄⁻ (Fig. 2d, e). In contrast, the introduction of DFOB⁻ resulted in the formation of a new Ca²⁺−DFOB⁻ ion pair, which significantly improved the distribution of Ca²⁺, as observed in the MD simulation snapshots (Fig. 2f, g). This was

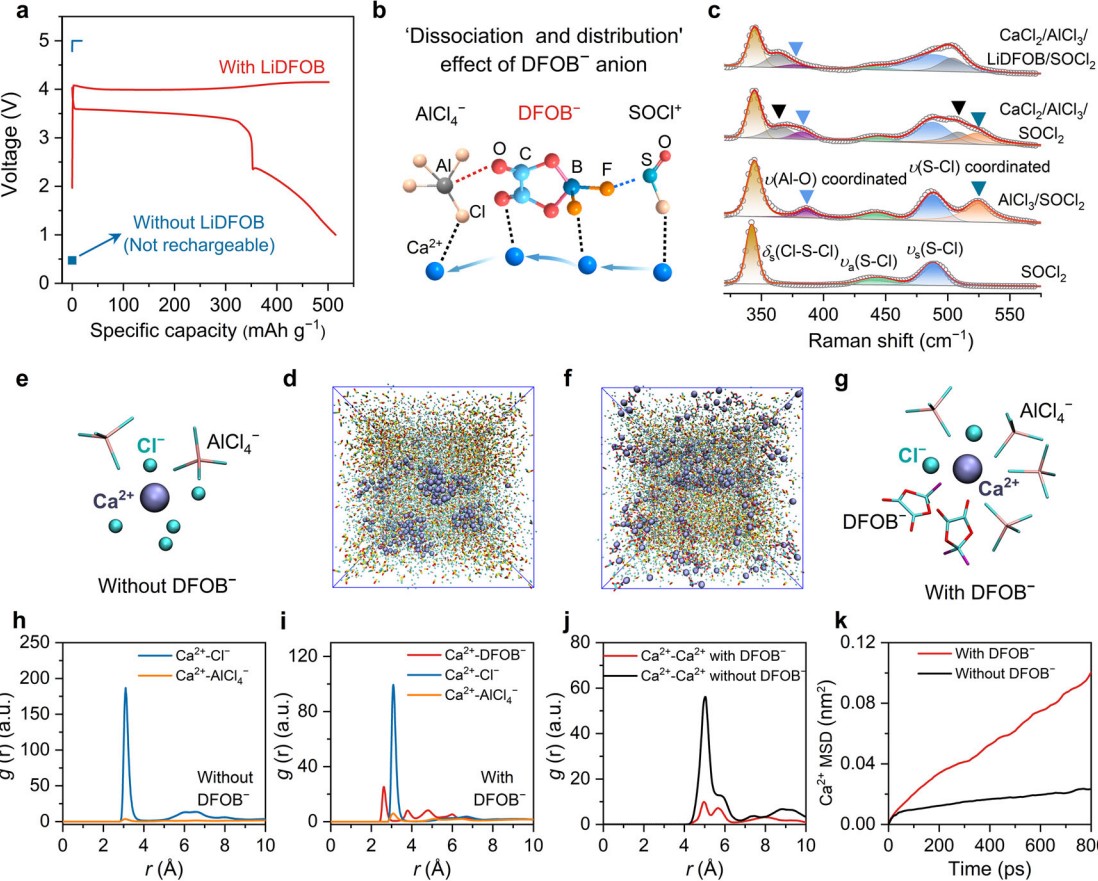

**Fig. 2 | "Dissociation−distribution" effect of LiDFOB in a CALS electrolyte.**
**a** Galvanostatic charge−discharge curves of rechargeable Ca/Cl₂ batteries using a CALS electrolyte with and without LiDFOB. The specific charge capacity and current density are 500 mAh g⁻¹ and 100 mA g⁻¹, respectively. **b** Schematic illustration of the 'dissociation−distribution' effect; i.e., the DFOB⁻ anion can facilitate the dissociation of various ions, such as AlCl₄⁻, SOCl⁺, and Ca²⁺, and benefit the distribution of Ca²⁺ to achieve fast migration and improve the kinetics. **c** Raman spectra of SOCl₂-

based electrolytes with different compositions. The marks in black color represent Ca−O vibrations from Ca²⁺···SOCl₂ interaction. **d, e** MD simulations and the corresponding solvation structure of the CALS electrolyte without LiDFOB, respectively. **f, g** MD simulations and the corresponding solvation structure of the CALS electrolyte, respectively. **h, i** RDFs of the main ion pairs in the CALS electrolyte without and with LiDFOB, respectively. **j, k** RDFs of the Ca²⁺−Ca²⁺ ion pair and MSD calculations of Ca²⁺ in the CALS electrolyte with and without LiDFOB, respectively.

attributed to C=O···Ca²⁺ and B–F···Ca²⁺ coordination originating from the unique chemical structure of DFOB⁻ (Fig. 2g and Supplementary Fig. 8). Radial distribution functions (RDFs) further confirmed the role of DFOB⁻ in the solvation structure. The Ca²⁺–DFOB⁻ ion pair distance (2.63 Å) was smaller than that of Ca²⁺–Cl⁻ (3.62 Å) (Fig. 2h, i), indicating that DFOB⁻ participated in a stronger interaction with Ca²⁺ that could compete with the interaction between Ca²⁺ and Cl-based species, thus improving their distribution (Fig. 2j). In addition, the interactions of AlCl₄⁻–SOCl⁺ and Cl⁻–Cl⁻ were both weakened in the presence of DFOB⁻ (Supplementary Figs 9 and 10), which was in good agreement with the decreased intensity of the Al–O and S–Cl vibration peaks in the Raman spectra (Fig. 2c). Mean square displacement (MSD) calculations showed that the Ca²⁺ migration rate increased 5-fold after the introduction of DFOB⁻ ($1.21 \times 10^{-6}$ vs. $0.24 \times 10^{-6}$ cm² s⁻¹) (Fig. 2k), which was consistent with the trend of ion diffusion coefficient in electrochemical impedance spectroscopy (EIS) (Supplementary Fig. 11). Notably, the possibility that Li ions were responsible for the increased conductivity or decreased impedance could be ruled out, based on the comparison of ionic conductivity and Nyquist plot based on the CaCl₂/AlCl₃/SOCl₂ electrolyte with and without LiCl (see details in Supplementary Fig. 11). Therefore, the improved Ca²⁺ distribution and ion dissociation mediated by DFOB⁻ were verified to benefit the rechargeability of our Ca/Cl₂ batteries.

Understanding cathode chemistry has important implications for improving battery performance. We thus analyzed the cathode reaction products at a variety of charge/discharge states (Fig. 3a–c). After the first discharge of the battery, the formation of CaCl₂, SO₂, and S was verified by X-ray photoelectron spectroscopy (XPS), consistent with Eq. (1) (Supplementary Fig. 12). The subsequent charge step, which corresponded to the conversion from CaCl₂ to Cl₂, exposed the edges of the graphite flakes (Fig. 3b and Supplementary Fig. 13), and the following discharge resulted in the formation of CaCl₂ nanoflakes on the graphite (Fig. 3c). The high-resolution transmission electron microscopy (HRTEM) and selected area electron diffraction (SAED) profiles further confirmed the formation of CaCl₂ after the first discharge (Fig. 3d). X-ray diffraction (XRD) also revealed the formation and removal of CaCl₂ on graphite at the fully discharged and charged states, respectively (Fig. 3e). An air-isolating chamber was used to avoid hygroscopicity of CaCl₂, and the weak CaCl₂ peaks could be still observed at the fully charged state because not all the CaCl₂ on cathode was consumed due to a limited charge capacity of 500 mAh g⁻¹. We suppose that Li⁺ does not play a critical role in the rechargeability of Ca/Cl₂ batteries, as verified by the battery performance based on the CaCl₂/AlCl₃/SOCl₂ electrolyte with the addition of LiCl, which could not deliver any charge/discharge capacity or plateau (Supplementary Fig. 14).

We next performed time-of-flight secondary ion mass spectrometry (TOF-SIMS) to visualize the three-dimensional distribution of CaCl₂ and Cl₂ on graphite at the fully discharged and charged states, and the results were in good agreement with the cathode reaction in Eq. (1)[27] (Fig. 3f, g). The formation and consumption of Cl₂ during battery charge and discharge were detected by DEMS (Fig. 3h). The high-resolution Cl 2p XPS spectra further confirmed the formation of Cl₂/C–Cl bond at the fully charged cathode[28,29] (Fig. 3i, j). In addition, the Ca/Cl₂ pouch cell showed no significant volume change at the fully charged state, which suggested that the generated Cl₂ might be adsorbed/trapped within the graphite cathode (Supplementary Fig. 15). These systematic characterizations of the cathode products confirmed the reversibility of the CaCl₂/Cl₂ redox reaction in our rechargeable Ca/Cl₂ batteries.

Our CALS electrolyte also improves the electrochemical stability of the Ca metal anode, which is important for battery rechargeability[30,31]. For instance, Ca metal that was immersed in CALS electrolyte without DFOB⁻ became dark immediately, whereas the surface remained shiny in CALS electrolyte with DFOB⁻ (Supplementary Fig. 16a, b). The XPS profile also showed metallic Al on Ca in the electrolyte without DFOB⁻, indicating that DFOB⁻ could suppress the parasitic reaction between AlCl₄⁻ and Ca metal, thus preventing the replacement of Al metal that inhibited Ca²⁺ transfer across the electrode/electrolyte interface (Supplementary Fig. 16c). The high-resolution F 1s XPS spectra of the Ca metal anode after the first discharge showed a B-F peak at ~686.6 eV, which was preserved after 40 cycles (Supplementary Fig. 17). It could be assigned to DFOB⁻ which was adsorbed on the surface of Ca metal, and its strong interaction with Ca²⁺ could weaken the interaction between Ca²⁺ and Cl⁻, which suppressed the parasitic chlorination of the Ca metal anode, thus promoting the electrochemical reversibility (Supplementary Fig. 18). Additionally, DFOB⁻ decomposition and Li deposition on the Ca metal anode might not occur, as suggested by the B 1s and Li 1s spectra (Supplementary Fig. 17).

We further verified the reversible morphological evolution of the Ca metal anode at a variety of charge/discharge states, without observation of 'dead' or dendritic Ca metal (Supplementary Figs. 19 and 20). The formation of CaCl₂ on the surface of the Ca metal anode was verified by the Ca 2p and Cl 2p XPS spectra (Supplementary Fig. 16). In a Ca/Au half cell containing a CALS electrolyte, the reversibility of Ca metal plating and stripping was determined to be 90.2% by cyclic voltammetry (Supplementary Fig. 21a). In contrast, in the CALS electrolyte without DFOB⁻, no stripping behavior was observed (Supplementary Fig. 21b), which confirmed the critical role of DFOB⁻ in the rechargeability of the Ca/Cl₂ battery. The scanning electron microscopy (SEM) and XRD results showed the uniform deposition of Ca metal on the Au foil in the presence of our CALS electrolyte (Supplementary Fig. 22). These results suggested that the parasitic reaction between the Ca metal anode and the electrolyte was suppressed, as mediated by LiDFOB, which is critical for the rechargeability of our Ca/Cl₂ battery.

We further investigated the electrochemical performance of our rechargeable Ca/Cl₂ battery. It demonstrated high electrochemical reversibility over a variety of specific capacities ranging from 200 to 1000 mAh g⁻¹ (Fig. 4a), as well as an impressive rate capability of 500 mA g⁻¹ (Fig. 4b), demonstrating the fast kinetics of the CaCl₂/Cl₂ redox reaction. We also verified the high cycling stability of our rechargeable Ca/Cl₂ battery over 100 cycles, whereas no rechargeability was observed when using the LiDFOB-free electrolyte (Fig. 4c). We sought to translate the impressive electrochemical performance into practical Ca metal batteries. For instance, retention performance (shelf life) is a critical parameter that determines the practical use of a battery[32]. To the best of our knowledge, however, current Ca metal batteries cannot afford sufficient retention capability, owing to the poor electrochemical stability of both electrodes used in conventional Ca-based electrolytes[33]. Our Ca/Cl₂ battery demonstrated remarkable retention capability, e.g., when maintained in a fully charged state for 1, 3, and 5 days, the rechargeability was well retained (Fig. 4d), indicating the high electrochemical stability of the charging product in our Ca/Cl₂ battery. For an as-prepared battery held at the open-circuit voltage for 30 days, 96.5% of the original specific capacity was retained, and highly consistent charge–discharge profiles and impedances were observed (Fig. 4e and Supplementary Fig. 23). These results suggested that our Ca/Cl₂ battery exhibited excellent retention performance in various charge/discharge states, which is highly attractive for practical applications.

The solid-state CALS electrolyte enables the production of separator-free Ca/Cl₂ batteries, which can effectively reduce the volume and weight of batteries for practical applications (Fig. 4f). In addition, the high ionic conductivities of our CALS electrolyte (e.g., 3.6 and 5.3 mS cm⁻¹ at 0 and 25 °C, respectively) allow us to explore low-temperature battery performance, which remains a major challenge for current Ca metal batteries. Remarkably, our Ca/Cl₂ battery functioned normally at 0 °C, retaining ~92.2% of its room-temperature

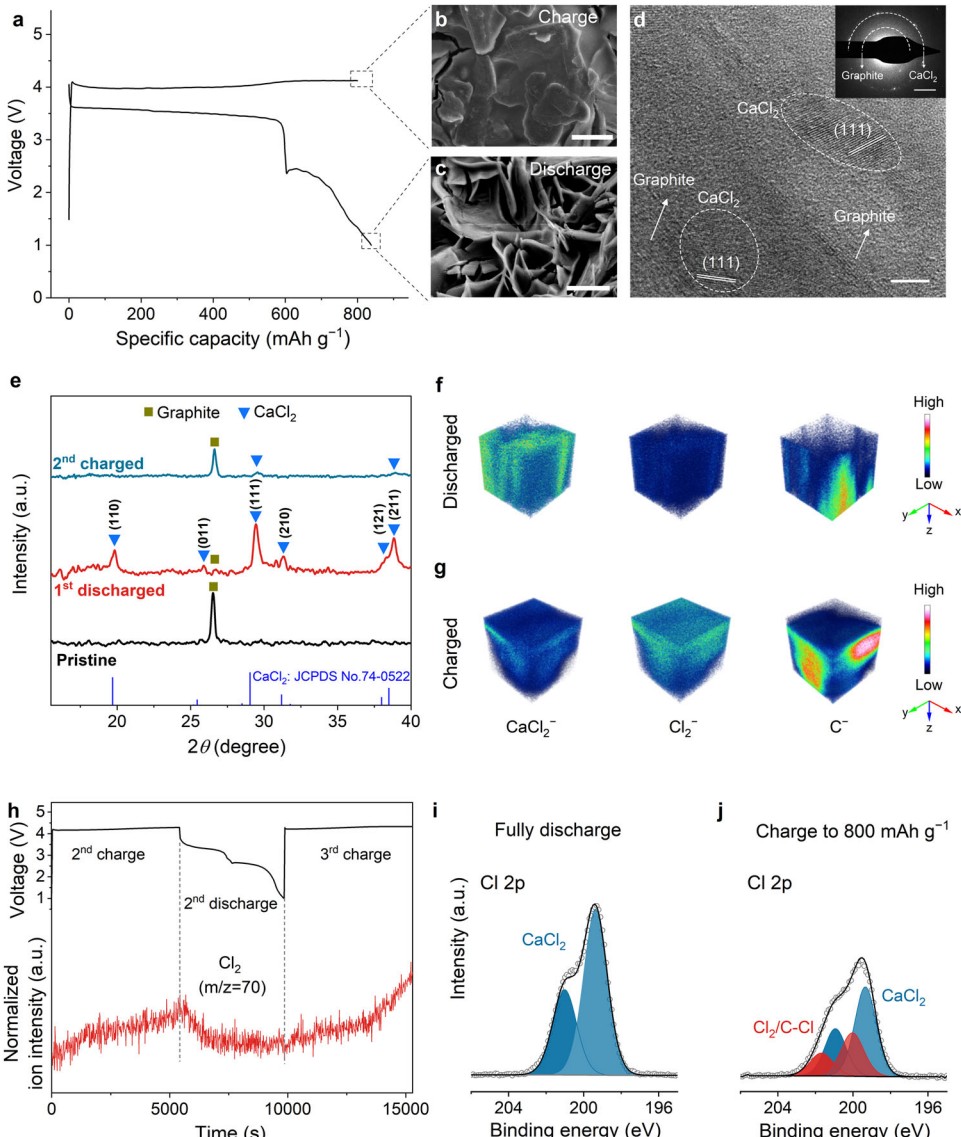

**Fig. 3 | Cathode product analysis of rechargeable Ca/Cl₂ batteries.**
**a** Galvanostatic charge–discharge curves of a rechargeable Ca/Cl₂ battery using the CALS electrolyte. **b**, **c** SEM images of the graphite cathode at the fully charged and discharged states, respectively. Scale bars, 1 μm. The current density and specific charge capacity are 100 mA g⁻¹ and 800 mAh g⁻¹, respectively. **d** HRTEM image of the discharge products formed on the graphite cathode after the first discharge. Scale bar, 5 nm. The inset shows the SAED patterns of CaCl₂. Scale bar of the inset, 2 nm⁻¹. **e** XRD patterns of the pristine, the first discharged, and the second charged graphite cathodes, respectively. **f**, **g** Depth distributions of CaCl₂⁻, Cl₂⁻, and C⁻ secondary ion fragments derived from TOF-SIMS depth scans of the fully discharged and charged graphite cathodes, respectively. Analysis areas, 50 × 50 μm². The current density and specific charge capacity in **e**–**g** are 100 mA g⁻¹ and 500 mAh g⁻¹, respectively. **h** DEMS profile of the rechargeable Ca/Cl₂ battery during continuous charge and discharge. **i**, **j** High-resolution Cl 2p XPS spectra of the graphite cathodes at the fully discharged and charged states, respectively.

energy efficiency (Fig. 4g), thus revealing fast ion transfer and reaction kinetics even at low temperatures. In addition, the high cycling stability was well maintained at 0 °C (Supplementary Fig. 24). To the best of our knowledge, this is the first rechargeable Ca metal battery with acceptable low-temperature capability to date[11,34,35], which represents an important step toward practical application.

## Discussion

In conclusion, we report the rechargeable Ca/Cl₂ battery based on the reversible cathode redox reaction between CaCl₂ and Cl₂, which introduces a new paradigm for Ca metal batteries with comprehensively high energy densities and rate capabilities. LiDFOB, a key electrolyte mediator with unique B−F and C=O functional groups, can facilitate the dissociation and distribution of Ca²⁺ and Cl-based species, thus enabling reversible and fast CaCl₂/Cl₂ redox reaction. The

prepared rechargeable Ca/Cl₂ battery delivers discharge voltages of 3 V and remarkable specific capacity (up to 1000 mAh g⁻¹) and rate capability (up to 500 mA g⁻¹). The retention and low-temperature performance also overcome the key bottlenecks that have hindered practical applications of Ca metal batteries to date. Our findings can not only benefit Ca metal batteries by enabling highly desirable cathode chemistry but also, in a broader context, revive the once overlooked Cl-based cathode chemistry for multivalent metal batteries, and harness them for sustainable, low-cost, and high-performance energy storage.

## Methods
### Preparation of the electrodes and electrolytes
Graphite (99%, Aladdin) was used for cathode preparation. It was mixed with polytetrafluoroethylene (PTFE, 60% aqueous dispersion,

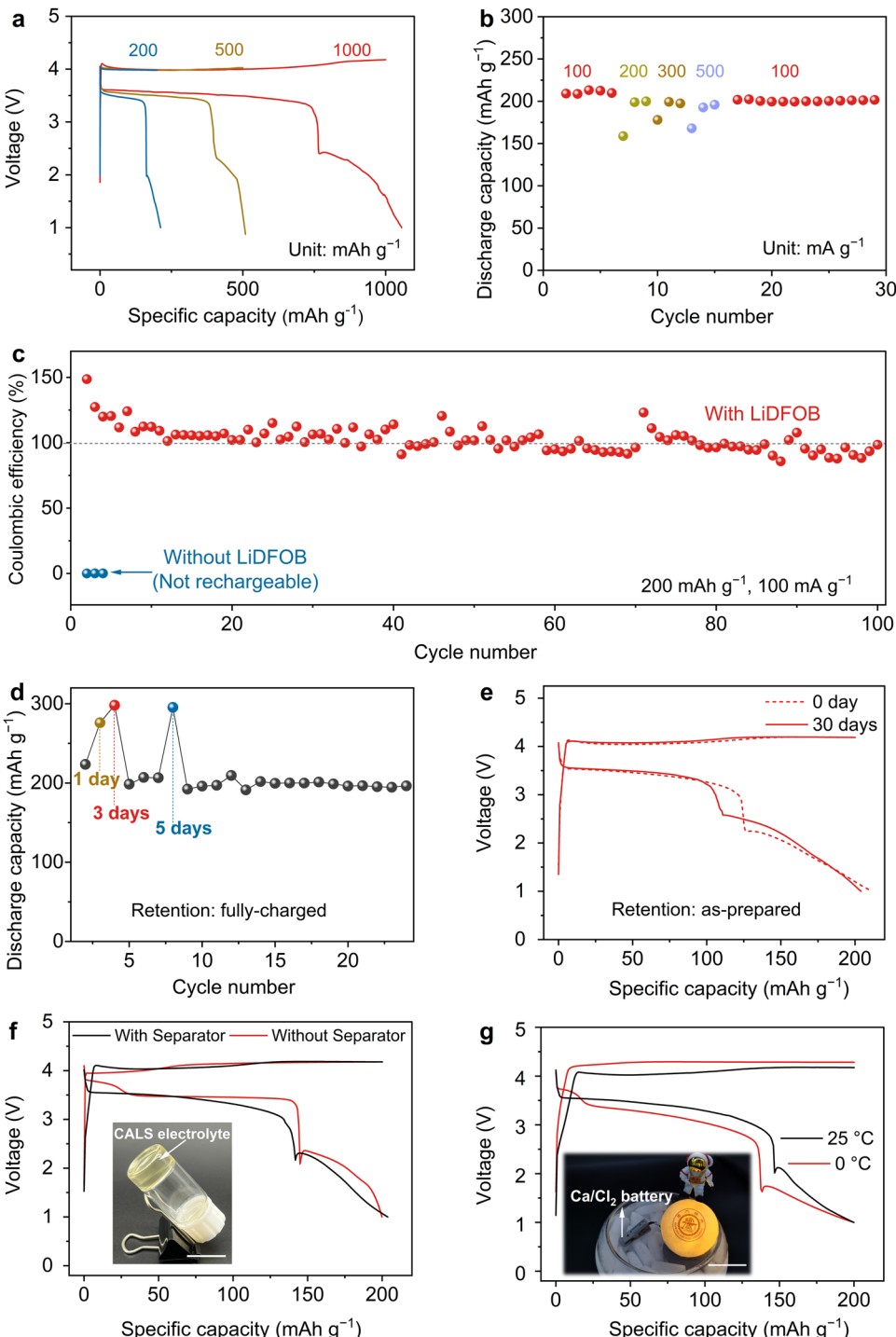

**Fig. 4 | Electrochemical performance of the rechargeable Ca/Cl₂ batteries.**
**a** Galvanostatic charge−discharge curves of a rechargeable Ca/Cl₂ battery using the CALS electrolyte with various specific charge capacities of 200, 500, and 1000 mAh g⁻¹. Current density, 100 mA g⁻¹. **b** Rate performance of the rechargeable Ca/Cl₂ battery with various current densities from 100 to 500 mA g⁻¹. Specific charge capacity, 200 mAh g⁻¹. **c** Cycling performance of rechargeable Ca/Cl₂ batteries using the CALS electrolyte with and without LiDFOB. **d** Electrochemical performance of the rechargeable Ca/Cl₂ battery with various retention durations of

1, 3, and 5 days. **e** Electrochemical performance of rechargeable Ca/Cl₂ batteries using the CALS electrolyte immediately and after 30 days. **f** Galvanostatic charge −discharge curves of rechargeable Ca/Cl₂ batteries with and without the use of a separator. The inset shows the solid-state CALS electrolyte in an inverted glass vial. Scale bar, 2 cm. **g** Galvanostatic charge−discharge curve of the rechargeable Ca/Cl₂ battery working at 0 °C. The inset shows a fully charged Ca/Cl₂ coin cell lighting up a commercial lamp at a low temperature. Scale bar, 5 cm. The specific charge capacity and current density in **c**−**g** are 200 mAh g⁻¹ and 100 mA g⁻¹, respectively.

Fuel Cell Store) at a mass ratio of 9:1 in ethanol solution. The obtained suspension was then subjected to ultrasonic treatment for 1 h to obtain the cathode slurry. A Ni foam was cut into a 14 mm diameter (1.54 cm²) circular using a manual disk cutter (MSK-T-10, MTI), followed by dropping 50 μL cathode slurry onto it. After evaporation of the ethanol

at 80 °C, the above slurry coating and drying processes were repeated until the mass loading of graphite reached 1.0−1.5 mg cm⁻². The obtained cathodes were further dried at 80 °C in a vacuum oven for 2 h. The Ca metal anodes were prepared by polishing the Ca flakes (99%, Sigma Aldrich, working area of 0.8 cm× 0.8 cm, thickness of

$0.5 \pm 0.04$ mm) in an argon-filled glovebox with the contents of $H_2O$ and $O_2$ below 1 ppm. All the electrolytes were also prepared in the same glovebox. $AlCl_3$ (99%, anhydrous, Aladdin) and $SOCl_2$ (99%, Aladdin) were used as received. LiDFOB (99%, anhydrous, Aladdin) and $CaCl_2$ (99.9%, anhydrous, Meryer) were dried at 120 °C for 12 h in a vacuum chamber before use. Typically, 6 M $AlCl_3$ and 1.3 M LiDFOB were dissolved in 1 mL $SOCl_2$, followed by the addition of 1.2 M $CaCl_2$ under continuous stirring for 1 h to obtain the CALS electrolyte. The gelation of the CALS electrolyte generally took 2 h, which allowed us to assemble batteries using the electrolyte in the liquid state.

## Electrochemical measurements

All the batteries were made inside an argon-filled glovebox with the contents of $H_2O$ and $O_2$ below 1 ppm. The prepared cathode and Ca metal anode were separated by one piece of glass fiber membrane (GF/D, Whatman, 16 mm in diameter) as the separator and 150 μL electrolyte was added to each coin cell (CR2032). All the electrochemical measurements were performed at 25 °C unless otherwise stated. The charge–discharge performance of the batteries was characterized using Neware battery testing systems (CT-4008-5V50mA-164-U). Electrochemical impedance spectroscopy (EIS) was conducted on a CHI600E electrochemical workstation. The electrochemical impedance spectroscopy measurement was conducted in a CR2032 coin cell, using a graphite cathode as the working electrode, and a Ca metal foil as the counter and reference electrode. The frequency ranged from 0.1 to $10^5$ Hz with an amplitude of 5 mV.

## Characterization

XRD, SEM, TOF-SIMS, XPS, Raman, DEMS, TEM, TGA, DSC, and XAS were performed, and detailed information was provided in Supplementary Information.

## Molecular dynamics (MD) simulations

All MD simulations were conducted using the GROMACS 2019.3 to investigate the solvation structure of electrolytes. The detailed information was provided in Supplementary Information.

## Reporting summary

Further information on research design is available in the Nature Portfolio Reporting Summary linked to this article.

## Data availability

The data that support the plots within this paper and other findings of this study are available from the corresponding author upon request.

## Code availability

The codes that support the findings of this study are available from the corresponding author upon request.

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

## Acknowledgements

This work was supported by the National Natural Science Foundation of China (22209108) and the Fundamental Research Funds for the Central Universities (23X010301599). The authors acknowledge the 4B7A station at the Beijing Synchrotron Radiation Facility.

## Author contributions

H.S. conceived and designed the research project. S.G. and X.Z. performed electrode preparation, electrolyte synthesis, battery production, and electrochemical characterization. B.Y. and Y.W. performed Raman spectroscopy. S.G., Q.X., Z.O., S.W., L.Y., and M.L. performed scanning electron microscopy, transmission electron microscopy, X-ray photo-electron spectroscopy, time-of-flight secondary ion mass spectrometry, thermogravimetric analysis, and differential scanning calorimetry. L.W. and S.G. analyzed molecular dynamics calculations. Yo. W., C.M., and X.J.Z performed Ca K-edge X-ray absorption experiments and Athena analysis. H.S., S.G., and X.Z. prepared the manuscript. All the authors participated in the data analysis and discussion.

## Competing interests

The authors declare no competing interests.
