## [Peer Review File · Nature Communications]

A rechargeable Ca/Cl₂ batteryREVIEWER COMMENTS

Reviewer #1 (Remarks to the Author):

In this manuscript, the authors reported a rechargeable Ca/Cl₂ battery with superior performance with supporting evidence from both the experimental and simulation sides. Such secondary calcium batteries are becoming promising candidates to revolutionize energy storage in the post-lithium-ion era. While agreeing that this work should be published to provoke future research in the area, the solvation structures proposed by the MD calculations (Fig. 2e and Fig. 2g) should be supported by XAS experimental evidence.

Another few minor questions are:

- 1) For Fig. 3e, would the authors explain why the 2nd charged states still show a minor (122) peak?
- 2) The authors mentioned that the gelation of the CALS electrolyte took around 2h, and the battery was assembled while the electrolyte was still in the liquid state. Is there any experimental evidence to show that the solvation structure was not changed before and after the gelation?

Overall, this manuscript is a good work and should be of interest to the readership. I recommend the acceptance of the manuscript after the authors properly address the questions mentioned here. Thanks.

Reviewer #2 (Remarks to the Author):

This manuscript reports the development of a Cl-based electrolyte composed of CaCl₂, AlCl₃, and LiDFOB salts in SOCl₂ (CALS electrolyte) for Ca/Cl₂ batteries with a calcium metal anode and Cl₂ cathode. The authors show that the CALS electrolyte improves battery performance and calcium electrodeposition. However, the results are not yet conclusive, and there are several important questions that need to be addressed before the manuscript can be published in a high-quality journal of Nature Communications.

Specific comments:

1. Equation 1: The authors should clarify why CaCl₂ is formed in situ at the cathode during the discharge process. Reviewer understand that the reaction in equation 1 occurs on the surface of the Ca metal anode. However, why is CaCl₂ formed in situ at the cathode during this reaction? Since there is no Ca metal at the cathode, the Ca source must be supplied from the anode. In that case, it is understandable that the CaCl₂ formed at the anode migrates to the cathode and is deposited into the graphite. Equation 1 is an electrochemical reaction because the discharge process is mentioned, but it appears to be a chemical reaction. The reviewer believes that the readability would be improved if the authors drew elementary reaction equations for each electrode.
2. CaCl₂ deposition on graphite cathode: The authors should explain why CaCl₂, which is soluble, can exist stably without dissolving on the metal electrode surface.
3. Capacity reporting: All capacities in this manuscript are listed per weight of graphite in the cathode. This makes it impossible to judge whether the capacities are large or not, and therefore comparisons with other studies are not very meaningful. At the very least, the reviewer would like to see the weight of the graphite and a description of the capacity per unit area, so that comparisons can be made.
4. Role of lithium: The authors should clarify the role of lithium in the battery. Is it possible that LiCl, but not CaCl₂, is the active chemical species in the cathode? In fact, the concentration of Li is higher than that of Ca in the electrolyte (1.3M LiDFOB and 1.2M CaCl₂).

5. CaCl₂-2H₂O detection by XRD: The authors should discuss the implications of detecting CaCl₂-2H₂O by XRD, which indicates that a significant amount of H₂O is present in the battery system. The presence of H₂O could deactivate the Ca metal anode.
6. Cl₂ detection by XPS: The authors should provide a detailed description of the state in which Cl₂ is present in the battery. If Cl₂ is in the gas phase, it is likely to desorb from the cathode electrode under vacuum during XPS measurement.
7. Electrodeposition of Ca metal: The authors should discuss why the CALS electrolyte is so effective in the electrodeposition of Ca metal, given that SOCl₂ is known to react violently with alkali and alkaline earth metals.
8. Li deposition: The authors should provide more evidence to support their claim that Li deposition is not occurring. If the authors are going to claim the absence of Li, they should explain the XPS instrument environment in detail and show that lithium can be properly detected.
9. Gelation of CALS electrolyte: The authors should describe the mechanism of gelation of the CALS electrolyte.

Reviewer #3 (Remarks to the Author):

The manuscript by Geng et al. reports the rechargeable Ca/Cl₂ battery based on a Cl-based electrolyte mediated by LiDFOB. The development of such beyond Li-ion batteries is crucial for the advancement of high theoretical capacity devices in the future. The charge-discharge curve of the battery is excellent, and its reversible capacity is larger than the other Ca cathode batteries previously reported. In addition, the results of molecular dynamics (MD) simulations are consistent with the Raman spectra and electrochemical impedance spectroscopy, which supports their interpretation of the role of LiDFOB. For this content, the paper would be acceptable for publication in Nature Communications.

However, from theoretical point of view, some clarifications are necessary for the MD simulations. I would like the authors to address the following minor comments.

1. For the reproduction of calculation results, the authors should mention about the number of electrolyte ions in the simulation box, or stoichiometric ratio of their simulated electrolyte and resulted electrolyte density with NPT ensemble.
2. The radial distribution function of Ca²⁺ - DFOB⁻ is not sufficient because we cannot recognize which DFOB part (F or shoulder O or tailed O) preferred Ca²⁺. It is recommended to add the pair distribution functions of Ca²⁺ - F, Ca²⁺ - Os, and Ca²⁺ - Ot.
3. For Fig. 2k, the unit of mean square displacement should be "nm²". And it is desired to mention about the diffusion coefficient of Ca²⁺ from the MD simulations.

Reviewer #4 (Remarks to the Author):

Title: A rechargeable Ca/Cl₂ battery

Authors: Shitao Geng, Xiaoju Zhao, Qiuchen Xu, Bin Yuan, Yan Wang, Meng Liao, Lei Ye, Shuo Wang, Zhaofeng Ouyang, Liang Wu, Hao Sun

In this article, a rechargeable Ca/Cl₂ battery is newly developed and its performance is characterized

by a wide range of experimental and computational means. The battery is reported to have high operating voltage, energy and power capacity, durability, and low-temperature performance, which the authors mainly attribute to the unique function of the LiDFOB salt added in the electrolyte.

The methods of analysis applied to this study are comprehensive and thorough and the outcome appears to have significant technological and scientific importance. However, I find some of their method description and interpretation of results are ambiguous or inconsistent, as listed below. If the authors address these problems or questions adequately, I can recommend the publication of this paper.

1. [p.2, lines 31-32 of SI] It is stated that "6 M AlCl₃ and 1.3 M LiDFOB were dissolved in 1 mL SOCl₂, followed by addition of 1.2 M CaCl₂". Does this mean that the concentrations of AlCl₃, LiDFOB, and CaCl₂ are 6 M, 1.3 M, and 1.2 M, respectively, in the final electrolyte? What is the actual composition of the electrolyte?

2. [p. 4, Eq. (1)] The stoichiometry is wrong in this reaction. Perhaps, "Ca" should be replaced by "2Ca²⁺ + 4e⁻"? If "Ca" must appear on the left-hand side of this equation, what is the source of this elemental Ca on the graphite cathode?

3. [p. 4, Eq. (2)] This equation is not the same as the "Cathode reaction" shown in Fig. 1a. Either one or both of these equations should be rewritten as a complete reaction, possibly including electrons explicitly.

4. [p. 5, lines 116-121] From the Raman spectrum of the CaCl₂/AlCl₃/LiDFOB/SOCl₂ solution, the authors draw two conclusions: (i) redshift of Al-O stretch peak indicates "weakening of the interaction between AlCl₃ and SOCl₂ mediated by DFOB-", and (ii) disappearance of the S-Cl stretch (AlCl₃...SOCl₂) peak indicates "sufficient dissociation between AlCl₃ and SOCl₂ mediated by DFOB-". However, the Al-O stretch peak intensity clearly increases upon addition of LiDFOB (top two spectra of Fig. 2c), making "sufficient dissociation between AlCl₃ and SOCl₂" very unlikely. Please clarify.

5. [p. 5, line 130] I think it is more proper to replace the expression "the electrolyte solvation shell" with "the solvation shell of Ca²⁺ ions".

6. [Fig. 2g] It appears that both F and O atoms of the DFOB anions make contact with Ca²⁺ ions. Which pairwise interaction, F...Ca²⁺ or O...Ca²⁺, is stronger and what is the average atomic composition of the Ca²⁺ solvation shell?

7. [SI, p.3, line 75] Reference 3 (J. Chem. Phys. 126, 014101 (2007)) was cited for the use of NPT ensemble. However, Ref. 3 describes a method for canonical (constant NVT) simulations, not for constant pressure simulations. The authors should either drop the citation or provide correct reference.

8. [p. 5, lines 141-142] Here, "decreased intensity of the Al-O and S-Cl vibration peaks in the Raman spectra (Fig. 2c)" is mentioned. However, as I noted in #4 above, the purple-colored peak labeled "ν(Al-O) coordinated" appears to have higher amplitude in the presence of LiDFOB (top-most spectrum) than in the absence of it (second spectrum). Please clarify.

9. [p. 5, lines 144-145] Here, values of the "ion diffusion coefficient" are reported for the electrolyte with and without LiDFOB, as obtained from EIS. Is this diffusion coefficient a kind of effective quantity representing a given electrolyte system? Or, is it possible to calculate this separately for Ca²⁺ and Li⁺ ions? Is it possible to rule out the possibility that Li ions are responsible for the increased conductivity (or decreased impedance)? Please clarify.

10. [p. 3, lines 68-82 of SI] I'm afraid the description of the simulation system and method is

incomplete.

First, what are the chemical species and their composition in the simulated system? Both SOCl_2 and SOCl^+ are mentioned in line 70 of p 3, SI. What is the ratio of each species in the simulated system and how was it determined? Without this fundamental information, the MD simulation result cannot be deemed credible.

Second, how was the long-range electrostatic interaction calculated in the simulation?

Technical Responses to Reviewers

Reviewer #1:

Remarks to the Author: *In this manuscript, the authors reported a rechargeable Ca/Cl₂ battery with superior performance with supporting evidence from both the experimental and simulation sides. Such secondary calcium batteries are becoming promising candidates to revolutionize energy storage in the post-lithium-ion era. While agreeing that this work should be published to provoke future research in the area, the solvation structures proposed by the MD calculations (Fig. 2e and Fig. 2g) should be supported by XAS experimental evidence.*

Response: Thank you very much for the valuable and insightful comments, which are very helpful to further strengthen this work. The manuscript has been carefully revised according to your important suggestions. For your convenience, the main revisions are marked with a yellow background in the revised Manuscript.

As kindly suggested, we provide XAS experimental evidence to verify the solvation structure proposed by MD calculations. Please check new Supplementary Fig. 7 and related discussion at the first paragraph of Page 5 in the revised Manuscript.

1. Another few minor questions are: For Fig. 3e, would the authors explain why the 2nd charged states still show a minor (122) peak?

Response: As kindly suggested, we provide explanation to the minor (122) peak at the 2nd charged state. We also optimized XRD measurement by using an air-isolating chamber to avoid hygroscopicity of CaCl₂. Please check the updated Fig. 3e and supplemented explanation at the first paragraph of Page 6 in the revised Manuscript.

2. The authors mentioned that the gelation of the CALS electrolyte took around 2h, and the battery was assembled while the electrolyte was still in the liquid state. Is there any experimental evidence to show that the solvation structure was not changed before and after the gelation?

Response: Thank you very much for the valuable suggestion! As kindly suggested, we supplemented the Raman spectra of the CALS electrolyte before and after gelation,

which were highly consistent, indicating that the solvation structure was not changed before and after the gelation. Please check new Supplementary Fig. 5 and related discussion at the last paragraph of Page 4 in the revised Manuscript.

3. Overall, this manuscript is a good work and should be of interest to the readership. I recommend the acceptance of the manuscript after the authors properly address the questions mentioned here. Thanks.

Response: Thanks again for your positive comments and constructive suggestions!

Responses to Reviewer #2

Remarks to the Author: *This manuscript reports the development of a Cl-based electrolyte composed of CaCl₂, AlCl₃, and LiDFOB salts in SOCl₂ (CALs electrolyte) for Ca/Cl₂ batteries with a calcium metal anode and Cl₂ cathode. The authors show that the CALs electrolyte improves battery performance and calcium electrodeposition. However, the results are not yet conclusive, and there are several important questions that need to be addressed before the manuscript can be published in a high-quality journal of Nature Communications.*

Response: We greatly appreciate the suggestive comments from the reviewer! We have fully followed them to revise our manuscript very carefully. In particular, we endeavor to make the results more conclusive and address the important questions according to your constructive suggestions. For your convenience, the main revisions are marked with a yellow background in the revised Manuscript.

1. Equation 1: The authors should clarify why CaCl₂ is formed in situ at the cathode during the discharge process. Reviewer understand that the reaction in equation 1 occurs on the surface of the Ca metal anode. However, why is CaCl₂ formed in situ at the cathode during this reaction? Since there is no Ca metal at the cathode, the Ca source must be supplied from the anode. In that case, it is understandable that the CaCl₂ formed at the anode migrates to the cathode and is deposited into the graphite. Equation 1 is an electrochemical reaction because the discharge process is mentioned, but it appears to be a chemical reaction. The reviewer believes that the readability would be improved if the authors drew elementary reaction equations for each electrode.

Response: Thank you very much for the critical suggestion! To improve the readability as suggested, we drew the elementary reaction equations for each electrode in new Equations (1) and (2), as well as the overall battery discharge reaction as new Equation (3).

According to Equation (1), the reduction of SOCl₂ leads to the formation of S, SO₂, and Cl⁻ during initial battery discharge, and Cl⁻ combines with Ca²⁺ from the CALs

electrolyte to form CaCl_2 at the cathode. The overall battery discharge reaction described in Equation (3) involves an automatic chemical reaction because the Gibbs free energy is smaller than zero, corresponding to the discharge process with four-electrons charge transfer through the CALS electrolyte. Please check new Equations (1), (2), and (3) on Page 3 in the revised Manuscript.

2. CaCl_2 deposition on graphite cathode: The authors should explain why CaCl_2 , which is soluble, can exist stably without dissolving on the metal electrode surface.

Response: As kindly suggested, we have explained the reason for stable existence of CaCl_2 on cathode without dissolving into the electrolyte. Please check related discussion at the third paragraph of Page 3 in the revised Manuscript.

3. Capacity reporting: All capacities in this manuscript are listed per weight of graphite in the cathode. This makes it impossible to judge whether the capacities are large or not, and therefore comparisons with other studies are not very meaningful. At the very least, the reviewer would like to see the weight of the graphite and a description of the capacity per unit area, so that comparisons can be made.

Response: We highly appreciate the reviewer's important suggestion! The mass loading of graphite is 1.2 mg cm^{-2} , which corresponded to a maximum areal capacity of $\sim 1.2 \text{ mAh cm}^{-2}$ in this work. As suggested, we also make comparison on the areal capacities of Ca metal batteries based on different cathodes. Please check new Supplementary Fig. 2 in the revised Supplementary Information. Thanks a lot!

4. Role of lithium: The authors should clarify the role of lithium in the battery. Is it possible that LiCl , but not CaCl_2 , is the active chemical species in the cathode? In fact, the concentration of Li is higher than that of Ca in the electrolyte (1.3M LiDFOB and 1.2M CaCl_2).

Response: Thank you very much for the insightful comments! The formation of LiCl on cathode could be excluded based on the XRD and XPS results, which showed strong CaCl_2 signals without observable signals of LiCl (Fig. 3e and i). This might be attributed to the higher electrostatic force of Ca^{2+} compared with that of Li^+ , which results in kinetically favorable formation of CaCl_2 compared to LiCl . In addition, we

compared the battery performance based on the $\text{CaCl}_2/\text{AlCl}_3/\text{SOCl}_2$ electrolyte with the addition of LiDFOB or LiCl with an equal concentration of 1.3 M. The battery using LiCl-based electrolyte could not deliver any charge/discharge capacities or plateaus (new Supplementary Fig. 14). Therefore, we suppose that it is DFOB^- , rather than Li^+ , which plays a critical role in the rechargeability of Ca/Cl₂ batteries. Please check new Supplementary Fig. 14 and related discussion on the role of lithium at the first paragraph of Page 6 in the revised Manuscript.

5. $\text{CaCl}_2\cdot 2\text{H}_2\text{O}$ detection by XRD: The authors should discuss the implications of detecting $\text{CaCl}_2\cdot 2\text{H}_2\text{O}$ by XRD, which indicates that a significant amount of H_2O is present in the battery system. The presence of H_2O could deactivate the Ca metal anode.

Response: Thank you very much for the important reminder! The detection of $\text{CaCl}_2\cdot 2\text{H}_2\text{O}$ was due to the strong hygroscopicity of CaCl_2 during XRD test in air. To avoid any further confusion, we have optimized the XRD and TEM measurements by using an air-isolating chamber, which showed bare CaCl_2 peaks without the presence of $\text{CaCl}_2\cdot 2\text{H}_2\text{O}$. Please check the updated Fig. 3e, d and related discussion at the first paragraph of Page 6 in the revised Manuscript.

6. Cl_2 detection by XPS: The authors should provide a detailed description of the state in which Cl_2 is present in the battery. If Cl_2 is in the gas phase, it is likely to desorb from the cathode electrode under vacuum during XPS measurement.

Response: These insightful comments and questions draw our serious attentions. As kindly suggested, we provided a detailed description on the state of Cl_2 , which might be in the gas phase, but adsorbed/trapped in the carbon cathode. This could be verified by the peak at ~200.0 eV in XPS that was assigned to C-Cl, which could avoid the desorb from the cathode even under vacuum (Zhu *et al. Proc. Natl. Acad. Sci. U.S.A.* **2023**, 120, e2310903120). In addition, the Ca/Cl₂ pouch cell stopped at fully charged state showed no significant volume change (new Supplementary Fig. 15), suggesting that Cl_2 might be adsorbed/trapped within the cathode. Besides, a small portion of Cl_2 could be dissolved in the SOCl_2 -based electrolyte. We will further work on this issue to acquire in-depth understandings on the mechanism. Please check new Supplementary Fig. 15 and related discussion at the second paragraph of Page 6 in the revised Manuscript. Thanks a lot!

7. *Electrodeposition of Ca metal: The authors should discuss why the CALS electrolyte is so effective in the electrodeposition of Ca metal, given that SOCl₂ is known to react violently with alkali and alkaline earth metals.*

Response: As kindly suggested, we provide detailed discussion on the effectiveness of CALS electrolyte for Ca metal electrodeposition. Briefly, DFOB⁻ showed strong interaction with Ca²⁺ which could suppress the interaction between Ca²⁺ and Cl⁻, which suppressed the parasitic chlorination of the Ca metal anode, thus promoting the electrochemical reversibility with higher electrochemical reversibility. Please check the related discussion at the last paragraph of Page 6 and the first paragraph of Page 7 in the revised Manuscript.

8. *Li deposition: The authors should provide more evidence to support their claim that Li deposition is not occurring. If the authors are going to claim the absence of Li, they should explain the XPS instrument environment in detail and show that lithium can be properly detected.*

Response: We appreciate the important suggestion from the reviewer. As kindly suggested, we supplemented the XPS instrument environment in detail in Methods of Supplementary Information. An easy way to verify that Li can be properly detected is to probe the Ca metal anode from a fully charged cell with or without rinsing with SOCl₂. XPS Li 1s spectrum showed an obvious peak at 56.1 eV without SOCl₂ rinsing, which could be assigned to the residual LiDFOB, indicating that Li could be detected by the XPS instrument. In contrast, no Li signal had been observed for the Ca metal anode with SOCl₂ rinsing, indicating that Li deposition was not occurring during the charging process (**Fig. R1**).

Fig. R1. XPS survey spectra of calcium anodes from fully charged Ca/Cl₂ batteries.

Calcium anodes were dried naturally without rinsing and dried after rinsing, respectively.

9. Gelation of CALS electrolyte: The authors should describe the mechanism of gelation of the CALS electrolyte.

Response: As kindly suggested, we supplemented thermogravimetric analysis (TGA) of the CALS electrolyte, which excluded chemical bonding or reactions between the components in the electrolyte (new Supplementary Fig. 2a). Differential scanning calorimetry (DSC) profiles showed no heat change or phase transition of the CALS electrolyte, indicating that the CALS electrolyte was a physical mixture (new Supplementary Fig. 2b). Therefore, the gelation of the CALS electrolyte can be mainly related to the intermolecular interactions between different substances such as Ca^{2+} and DFOB^- (Zheng *et al. Nat. Commun.* **2019**, 10, 1604; Mayr *et al. Chem. Soc. Rev.* **2018**, 47, 1484-1515). Please check new Supplementary Fig. 6 and related discussion at the last paragraph of Page 4 in the revised Manuscript.

Responses to Reviewer #3

Remarks to the Author: *The manuscript by Geng et al. reports the rechargeable Ca/Cl₂ battery based on a Cl-based electrolyte mediated by LiDFOB. The development of such beyond Li-ion batteries is crucial for the advancement of high theoretical capacity devices in the future. The charge-discharge curve of the battery is excellent, and its reversible capacity is larger than the other Ca cathode batteries previously reported. In addition, the results of molecular dynamics (MD) simulations are consistent with the Raman spectra and electrochemical impedance spectroscopy, which supports their interpretation of the role of LiDFOB. For this content, the paper would be acceptable for publication in Nature Communications. However, from theoretical point of view, some clarifications are necessary for the MD simulations. I would like the authors to address the following minor comments.*

Response: We greatly appreciate the reviewer's positive and insightful comments, which are very helpful to improve our work. The manuscript has been carefully revised according to the valuable advices. The main revisions are marked with a yellow background in the revised manuscript.

1. For the reproduction of calculation results, the authors should mention about the number of electrolyte ions in the simulation box, or stoichiometric ratio of their simulated electrolyte and resulted electrolyte density with NPT ensemble.

Response: As kindly suggested, we have provided the number of electrolyte ions of the simulated electrolyte in Supplementary Table 1. Specifically, the CALS simulation box contains 1100 AlCl₄⁻, 238 DFOB⁻, 238 Li⁺, 220 Ca²⁺, 440 Cl⁻, 1100 SOCl⁺, and 1411 SOCl₂, respectively. As for CAS system, the ions and numbers were the same as CALS system except for the presence of DFOB⁻ and Li⁺ ions. The CALS and CAS electrolyte densities with NPT ensemble were supplemented to Molecular dynamics (MD) simulations in the Method section on Page 5 of the revised Supplementary Information. Thanks a lot!

2. The radial distribution function of Ca²⁺ – DFOB⁻ is not sufficient because we cannot recognize which DFOB part (F or shoulder O or tailed O) preferred Ca²⁺. It is recommended to add the pair distribution functions of Ca²⁺ – F, Ca²⁺ – Os, and Ca²⁺ – Ot.

Response: We greatly appreciate the important suggestion from the reviewer. As suggested, we have supplemented the RDFs of the pair distribution functions of Ca^{2+} -F, Ca^{2+} - O_s , and Ca^{2+} - O_t , and the interaction intensity follows the order of Ca^{2+} - $\text{O}_t > \text{Ca}^{2+}$ -F \gg Ca^{2+} - O_s . Please check new Supplementary Fig. 8 in the revised Supplementary Information.

3. For Fig. 2k, the unit of mean square displacement should be “nm²”. And it is desired to mention about the diffusion coefficient of Ca^{2+} from the MD simulations.

Response: Thank you very much for the important reminder! We have corrected the unit of mean square displacement in updated Fig. 2k. In addition, the diffusion coefficient of Ca^{2+} in the MD simulations has been provided. Please check the updated Fig. 2k and related discussion at the first paragraph of Page 5 in the revised Manuscript. Thanks again for your constructive comments and suggestions!

Responses to Reviewer #4

Remarks to the Author: *In this article, a rechargeable Ca/Cl₂ battery is newly developed and its performance is characterized by a wide range of experimental and computational means. The battery is reported to have high operating voltage, energy and power capacity, durability, and low-temperature performance, which the authors mainly attribute to the unique function of the LiDFOB salt added in the electrolyte. The methods of analysis applied to this study are comprehensive and thorough and the outcome appears to have significant technological and scientific importance. However, I find some of their method description and interpretation of results are ambiguous or inconsistent, as listed below. If the authors address these problems or questions adequately, I can recommend the publication of this paper.*

Response: Thank you very much for the positive comments! The manuscript has been carefully revised according to your constructive suggestions. In particular, we have further enhanced the method description and interpretation of results to avoid any further confusion. For your convenience, the main revisions are marked with a yellow background in the revised Manuscript.

1. [p.2, lines 31-32 of SI] *It is stated that "6 M AlCl₃ and 1.3 M LiDFOB were dissolved in 1 mL SOCl₂, followed by addition of 1.2 M CaCl₂". Does this mean that the concentrations of AlCl₃, LiDFOB, and CaCl₂ are 6 M, 1.3 M, and 1.2 M, respectively, in the final electrolyte? What is the actual composition of the electrolyte?*

Response: We appreciate the important question from the reviewer. The concentrations of AlCl₃, LiDFOB, and CaCl₂ in the CALS electrolyte are 6 M, 1.3 M, and 1.2 M, respectively. Thanks a lot!

2. [p. 4, Eq. (1)] *The stoichiometry is wrong in this reaction. Perhaps, "Ca" should be replaced by "2Ca²⁺ + 4e⁻"? If "Ca" must appear on the left-hand side of this equation, what is the source of this elemental Ca on the graphite cathode?*

Response: As kindly suggested, the stoichiometry has been revised in the revised Manuscript. According to new Eq. (1) in the revised Manuscript, the reduction of SOCl₂ leads to the formation of S, SO₂, and Cl⁻ during the initial discharge, and the Cl⁻ combines with Ca²⁺ from the CALS electrolyte, resulting in the formation of CaCl₂ at

the cathode. Please check new Eqs. (1), (2), and (3) and related discussion at the last paragraph of Page 3 in the revised Manuscript.

3. [p. 4, Eq. (2)] *This equation is not the same as the "Cathode reaction" shown in Fig. 1a. Either one or both of these equations should be rewritten as a complete reaction, possibly including electrons explicitly.*

Response: As kindly suggested, we have supplemented the complete reactions including electrons explicitly in new Eq. (4), (5), and (6) on Page 4 in the revised Manuscript. Thanks a lot!

4. [p. 5, lines 116-121] *From the Raman spectrum of the $\text{CaCl}_2/\text{AlCl}_3/\text{LiDFOB}/\text{SOCl}_2$ solution, the authors draw two conclusions: (i) redshift of Al-O stretch peak indicates "weakening of the interaction between AlCl_3 and SOCl_2 mediated by DFOB-", and (ii) disappearance of the S-Cl stretch ($\text{AlCl}_3\cdots\text{SOCl}_2$) peak indicates "sufficient dissociation between AlCl_3 and SOCl_2 mediated by DFOB-". However, the Al-O stretch peak intensity clearly increases upon addition of LiDFOB (top two spectra of Fig. 2c), making "sufficient dissociation between AlCl_3 and SOCl_2 " very unlikely. Please clarify.*

Response: These critical questions draw our serious attentions. Following your constructive suggestion, we have carefully analyzed and fitted the mentioned Raman peaks in updated Fig. 2c, which showed clear decrease and red shift of the Al-O stretching peak upon LiDFOB addition. We had mistakenly neglected the peak at 363.8 and 504.5 cm^{-1} which were supposed to originate from the interaction between Ca^{2+} and SOCl_2 in the original Manuscript. Please check updated Fig. 2c and related discussion at the last paragraph of Page 4 in the revised Manuscript. Thanks a lot!

5. [p. 5, line 130] *I think it is more proper to replace the expression "the electrolyte solvation shell" with "the solvation shell of Ca^{2+} ions".*

Response: As kindly suggested, we use 'the solvation shell of Ca^{2+} ions' in the revised Manuscript. Please check the first paragraph of Page 5 in the revised Manuscript. Thanks a lot!

6. [Fig. 2g] It appears that both F and O atoms of the DFOB anions make contact with Ca^{2+} ions. Which pairwise interaction, $\text{F}\dots\text{Ca}^{2+}$ or $\text{O}\dots\text{Ca}^{2+}$, is stronger and what is the average atomic composition of the Ca^{2+} solvation shell?

Response: We appreciate the important comment from the reviewer. The interaction of $\text{O}\dots\text{Ca}^{2+}$ is stronger than $\text{F}\dots\text{Ca}^{2+}$ according to the radial distribution functions from MD simulations in the new Supplementary Fig. 8. We also provide the average atomic composition of the Ca^{2+} solvation shell in new Supplementary Table 2.

7. [SI, p.3, line 75] Reference 3 (*J. Chem. Phys.* 126, 014101 (2007)) was cited for the use of NPT ensemble. However, Ref. 3 describes a method for canonical (constant NVT) simulations, not for constant pressure simulations. The authors should either drop the citation or provide correct reference.

Response: As suggested, new Reference 7 and 8 have been provided at Page 4 of the revised Supplementary Information. Thanks a lot!

8. [p. 5, lines 141-142] Here, "decreased intensity of the Al–O and S–Cl vibration peaks in the Raman spectra (Fig. 2c)" is mentioned. However, as I noted in #4 above, the purple-colored peak labeled " $\nu(\text{Al-O})$ coordinated" appears to have higher amplitude in the presence of LiDFOB (top-most spectrum) than in the absence of it (second spectrum). Please clarify.

Response: We appreciate the reviewer's important question. We have carefully analyzed and fitted the mentioned Raman spectrum in Fig. 2c, in which the intensity of the " $\nu(\text{Al-O})$ coordinated" peak was decreased upon LiDFOB addition. We are sorry for neglecting the peak at 363.8 and 504.5 cm^{-1} originated from the interaction between Ca^{2+} and SOCl_2 in the previous version. Please check updated Fig. 2c and related discussion at the last paragraph of Page 4 in the revised Manuscript. Thanks a lot!

9. [p. 5, lines 144-145] Here, values of the "ion diffusion coefficient" are reported for the electrolyte with and without LiDFOB, as obtained from EIS. Is this diffusion coefficient a kind of effective quantity representing a given electrolyte system? Or, is it possible to calculate this separately for Ca^{2+} and Li^+ ions? Is it possible to rule out the possibility that Li ions are responsible for the increased conductivity (or decreased

impedance)? Please clarify.

Response: We greatly appreciate the important comment from the reviewer! The ion diffusion coefficient from EIS is actually a kind of effective quantity representing a given electrolyte system, which involves the ion diffusion impedance inside the electrodes for a given electrolyte (Levi *et al. J. Phys. Chem. B* 1997, 101, 4630). As kindly suggested, we measured the ionic conductivities of CaCl₂/AlCl₃/SOCl₂ electrolyte with addition of 1.3 M LiDFOB or LiCl, which exhibited no significant increases in ionic conductivity (**Figure R2a**, as new Supplementary Fig. 11a), suggesting that the Li⁺ ions are not responsible for the increased conductivity.

To rule out the role of Li⁺ ion in impedance, we further supplemented EIS measurement using Ca metal/graphite cells based on CaCl₂/AlCl₃/SOCl₂ and CaCl₂/AlCl₃/SOCl₂/LiCl electrolytes (**Figure R2b, c**, as new Supplementary Fig. 11b, c). The addition of LiCl made negligible influence on R_{ct} and ion diffusion coefficient compared to the LiCl-free electrolyte, which ruled out the possibility of Li⁺ ions for the decreased impedance. Therefore, we suppose that the possibility that Li ions are responsible for the increased conductivity (or decreased impedance) could be ruled out. Please check relative discussions in the first paragraph of Page 5 of the revised Manuscript. Thanks again for your critical suggestions!

Figure R2. a, Ion conductivities of CaCl₂/AlCl₃/SOCl₂, LiCl/CaCl₂/AlCl₃/SOCl₂ and LiDFOB/CaCl₂/AlCl₃/SOCl₂ electrolytes. **b**, Nyquist plots of the as-prepared Ca/Cl₂ batteries using the CaCl₂/AlCl₃/SOCl₂ and LiCl/CaCl₂/AlCl₃/SOCl₂ electrolytes. The inset showed the corresponding equivalent circuit. **c**, Linear correlations between Z' and the square root of frequency ($\omega^{-1/2}$) in the low-frequency regions of the EIS plots using as-prepared Ca/Cl₂ batteries.

10. [p. 3, lines 68-82 of SI] I'm afraid the description of the simulation system and

method is incomplete. First, what are the chemical species and their composition in the simulated system? Both SOCl_2 and SOCl^+ are mentioned in line 70 of p 3, SI. What is the ratio of each species in the simulated system and how was it determined? Without this fundamental information, the MD simulation result cannot be deemed credible. Second, how was the long-range electrostatic interaction calculated in the simulation?

Response: We highly appreciate your valuable comments and suggestions on the description of the simulation system and method. The chemical species in the simulated system were AlCl_4^- , DFOB^- , Li^+ , Ca^{2+} , Cl^- , SOCl^+ , SOCl_2 , and their composition was determined by the actual concentration of each species in the electrolyte. In particular, AlCl_3 could react with equal molar amount of SOCl_2 to form SOCl^+ and AlCl_4^- according to the following equation (Mosier-Boss *et al. J. Chem. Soc., Faraday Trans. 1.* 1989, 85, 11-21), which resulted in the coexistence of SOCl_2 and SOCl^+ in our simulated system:

Therefore, the electrolyte composition of the 6 M AlCl_3 , 1.2 M CaCl_2 and 1.3 M LiDFOB in SOCl_2 electrolyte was 6 M AlCl_4^- , 1.2 M Ca^{2+} , 2.4 M Cl^- , 1.3 M Li^+ , 1.3 M DFOB^- , 7.7 M SOCl_2 and 6 M SOCl^+ , corresponding to 1100 AlCl_4^- , 238 DFOB^- , 238 Li^+ , 220 Ca^{2+} , 440 Cl^- , 1100 SOCl^+ , and 1411 SOCl_2 , respectively in the simulation box. To avoid any further confusion, we have supplemented the above-mentioned discussion in updated Supplementary Table 1 in the revised Supplementary Information.

The long-range electrostatic interaction was calculated based on the Particle-mesh Ewald (PME) method. It used Ewald summation to split up the calculation into a short-range part, for which all interactions are directly evaluated up to a cutoff radius r_c (1.2 nm), and a long-range part, which is solved in a reciprocal space. To take advantage of fast Fourier transforms (FFTs) for the conversions to reciprocal space, the charges are interpolated onto a uniform grid using cardinal B-splines. The advantage of this method is the rapid convergence of the energy compared with that of a direct summation, thus enabling high accuracy and reasonable speed when computing long-range interactions, making it a standard method for calculating long-range interactions in periodic systems. Please check the related explanations on Page 5 of the revised Supplementary Information. Thank you very much for the valuable suggestions.

REVIEWERS' COMMENTS

Reviewer #1 (Remarks to the Author):

I would recommend the acceptance of the manuscript in its current version.

Reviewer #2 (Remarks to the Author):

The authors have made suitable revisions. The revised version is recommended for publication in Nature Communications.

Reviewer #3 (Remarks to the Author):

The authors have carefully revised the manuscript and addressed all my comments. Now, the current manuscript is suitable for publication in Nature Communications.

Reviewer #4 (Remarks to the Author):

I find that all of my previous comments have been adequately addressed and reflected in the revised manuscript. I'd like to recommend that this paper be accepted for publication.

Technical Responses to Reviewers

Reviewer #1:

Remarks to the Author: I would recommend the acceptance of the manuscript in its current version.

Response: Thanks a lot!

Reviewer #2:

Remarks to the Author: The authors have made suitable revisions. The revised version is recommended for publication in Nature Communications.

Response: Thank you!

Reviewer #3:

Remarks to the Author: The authors have carefully revised the manuscript and addressed all my comments. Now, the current manuscript is suitable for publication in Nature Communications.

Response: Thanks a lot!

Reviewer #4:

Remarks to the Author: I find that all of my previous comments have been adequately addressed and reflected in the revised manuscript. I'd like to recommend that this paper be accepted for publication.

Response: Thank you!